# PROGRAMMATIC CONTEXT AUGMENTATION FOR LLM-BASED SYMBOLIC REGRESSION

## ABSTRACT

Symbolic regression (SR), the task of discovering mathematical expressions that best describe a given dataset, remains a fundamental challenge in scientific discovery. Traditional approaches, primarily based on genetic algorithms and related evolutionary methods, have proven useful but suffer from scalability and expressivity limitations. Recently, large language model (LLM)-based evolutionary search methods have been introduced into SR and show promise. However, existing LLM-based approaches typically rely on scalar evaluation metrics, such as mean squared error, as the sole source of feedback during the search process, thereby overlooking the rich information embedded in the dataset. To address this limitation, we propose a novel LLM-based evolutionary search framework that incorporates programmatic context augmentation. By enabling code-based interactions with the dataset, our method can actively perform data analysis and extract informative signals, beyond aggregated evaluation scores. We evaluate our framework on advanced benchmarks, such as LLM-SRBench, and demonstrate superior efficiency and accuracy compared to strong baselines.

## 1 INTRODUCTION

Symbolic regression (SR) (Kronberger et al., 2024; Makke & Chawla, 2024) is a central task across many scientific domains. Given a dataset collected from observations or experiments, the goal of SR is to discover concise and interpretable mathematical equations that can best explain the underlying structure of the data. Unlike purely predictive models, SR offers human-readable equations that provide interpretability and scientific insight, making it a powerful tool for discovery.

The key challenges of SR lie in efficiently searching the vast combinatorial space of candidate equations. Traditional approaches focus on using genetic programming (Kronberger et al., 2024), which iteratively generate populations of equations, evaluate them using fitness scores derived from the data, and refine them through mutation and crossover. Other approaches include neural-guided search (Cranmer et al., 2020; Shah et al., 2020) and reinforcement learning (Petersen et al., 2019). While these methods have shown success in small-scale settings, they remain limited in efficiency and scalability, largely due to their reliance on random mutations and their inability to leverage prior experience. To improve upon these limitations, neural network–based SR methods trained on datasets of equation–data pairs have been proposed (Kamienny et al., 2022).

More recently, large language models (LLMs) have been introduced into SR through evolutionary search frameworks (Shojaee et al., 2024). At each iteration, a textual prompt that combines the problem description with insights from past iterations is provided to the LLM, which then generates new candidate equations. This paradigm leverages the domain knowledge encoded in pretrained LLMs and has demonstrated promising results. Methods such as concept-library learning (Grayeli et al., 2024) have further enhanced LLMs' ability to reason symbolically in SR.

Despite these advances, current LLM-based SR methods remain constrained by their feedback mechanisms. Specifically, they rely almost exclusively on scalar evaluation metrics—most commonly mean squared error (MSE)—as the sole interaction with the dataset during the search process. That is, while candidate equations are evaluated against the dataset, the only signal returned to the model is a single numerical score. This stands in stark contrast to how human scientists approach the same task: rather than relying only on scalar evaluations, they actively conduct exploratory data

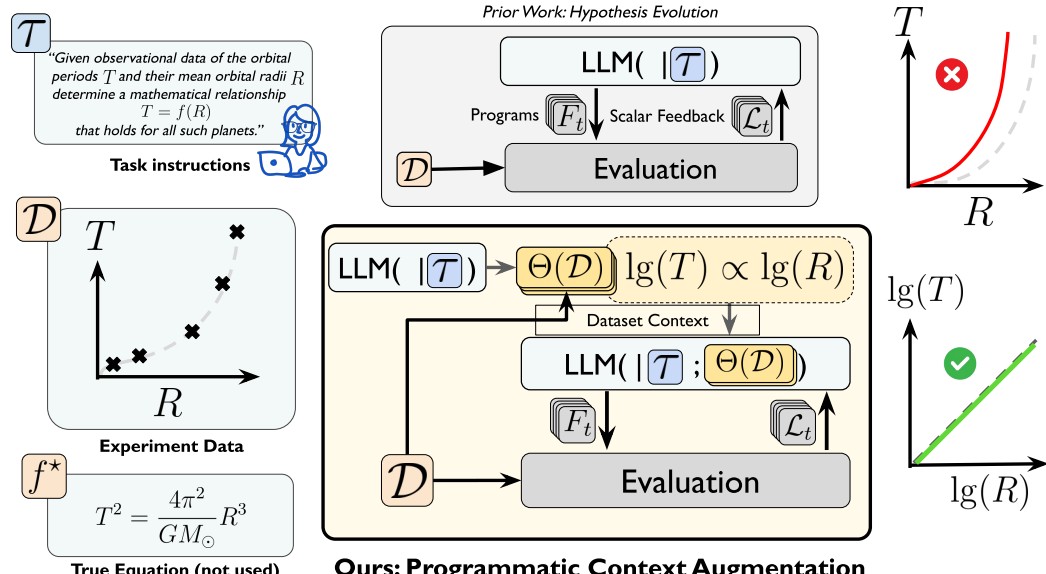

Figure 1: An overview of PROAUG, with $\tau$ being the task instructions, $D$ being the dataset, $\Theta$ being the dataset context, $F_t$ being the generated program and $L_t$ being the scalar feedback. Prior work in LLM guided symbolic regression (Shojaee et al., 2024) (in gray) rely on simplistic scalar evaluation metrics, such as MSE as feedback, ignoring rich information contained in dataset. Instead, PROAUG enriches the model's interaction with data by assigning the LLM a dual role-to generate data analysis code and extract informative signals for context augmentation. Considering the classical problem of deriving Kepler's third law from planetary motion data as a example. The law states that the square of a planet's orbital period ($T$) is proportional to the cube of its semi-major axis ($R$), i.e., $T^2 \propto R^3$. While standard method might struggle to identify this nonlinear relationship directly, PROAUG leverages the LLM's ability to reason by data anaysis. Specifically, the LLM generates code to apply a logarithmic transformation to both $T$ and $R$, yielding $\log(T)$ and $\log(R)$. When plotted, these transformed variables exhibit a clear linear relationship: $\log(T) \approx \frac{3}{2} \log(R) + \text{constant}$. This linear trend immediately suggests a power-law relationship between the original variables. Our method demonstrates how data-driven statistics complement structural constraints in equation discovery.

analysis, examining relationships, distributions, and trends to guide the design of hypothesis equations. In this work, we bridge this gap by introducing **PROgrammatic context AUGmentation** (PROAUG). Our framework empowers the LLM not only to propose candidate equations but also to generate code for analyzing the underlying dataset. By executing such code, the model gains access to richer signals—such as statistical properties and variable correlations—that can be used to refine its evolutionary search.

Concretely, we propose a new LLM-based SR framework in which the model is tasked with two complementary objectives: (1) proposing potential equation candidates, and (2) generating data-analysis code that extracts informative signals from the dataset. To rigorously assess our method, we benchmark it on LLM-SRBench (Shojaee et al., 2025), a comprehensive and recently proposed evaluation suite that mitigates concerns about LLMs leveraging memorized knowledge. Across both zero-shot and supervised fine-tuning settings, our method consistently outperforms strong baselines in terms of both efficiency and accuracy.

Our contributions are summarized as follows:

- We introduce **programmatic context augmentation**, a novel mechanism that enables LLMs to actively interact with the dataset during evolutionary SR.

- We design a dual-task framework in which the LLM generates both candidate equations and code for data-driven analysis.

- We evaluate our approach on LLM-SRBench and demonstrate consistent improvements over state-of-the-art baselines across multiple settings. For instance, we see a 3x error reduction on LSR-Transform when using programmatic context augmention with DeepSeek-V3.1 as the base model.

## 2 PRELIMINARIES

**Symbolic Regression.** Using the framework of Empirical Risk Minimization (ERM), the problem of symbolic regression can be formalized as follows (Kronberger et al., 2024; Vapnik, 1991).

Given a dataset $\mathcal{D} = \{(\mathbf{x}_i, y_i)\}_{i=1}^n$, where $\mathbf{x}_i \in \mathbb{R}^d$ denotes the input features and $y_i \in \mathbb{R}$ the scalar target, the objective of SR is to find a function $f^* \in \mathcal{F}$ that minimizes the empirical loss:

$$f^* = \arg\min_{f \in \mathcal{F}} \mathcal{L}(f),$$

where $\mathcal{F}$ is the discrete hypothesis class of functions mapping $\mathbb{R}^d$ to $\mathbb{R}$, and $\mathcal{L}$ is a loss function measuring predictive accuracy. A common choice is the mean squared error (MSE):

$$\mathcal{L}(f) = \frac{1}{n} \sum_{i=1}^n \left(y_i - f(\mathbf{x}_i)\right)^2.$$

Since the hypothesis class $\mathcal{F}$ is constrained by Python code, one can also view this problem as an instance of program synthesis (Chaudhuri et al., 2021).

**LLM evolutionary search for SR.** Recent work has integrated large language models (LLMs) into the framework of evolutionary algorithms, yielding LLM agents that iteratively propose solutions to tasks such as coding or scientific discovery. We focus on their application to SR, as exemplified by LLM-SR (Shojaee et al., 2024), which serves as a main baseline in this work. The framework consists of three key components:

*(i) Hypothesis generation.* At each iteration, the LLM is prompted with a structured input that includes: task instructions, problem specifications (e.g., the physical meaning of variables), the optmization and evaluation function, and demonstrations of promising prior hypotheses and their improvement trajectories from an experience buffer. Conditioned on this prompt, the LLM generates a candidate equation in the form of a Python function skeleton with learnable parameters.

*(ii) Hypothesis optimization and assessment.* The free parameters of the generated equation skeletons are optimized using external solvers (e.g., NumPy with BFGS). The resulting hypothesis is then evaluated on the dataset to produce a fitness score. In LLM-SR, this score is defined as the negative MSE on the training set. However, we observed that this definition can lead to generalization mismatches between training and test performance. To mitigate this, we split the training set into a *train-train* (tr-tr) and *train-val* (tr-val) subset. Parameters are optimized on the train-train set, and the fitness score is computed on the train-val set using the negative normalized MSE.

*(iii) Experience management.* Inspired by mechanisms in evolutionary algorithms (e.g., island models (Cranmer, 2023; Romera-Paredes et al., 2024)), LLM-SR maintains an experience buffer of past hypotheses. Candidate hypotheses are sampled from this buffer according to a distribution weighted by fitness scores, and incorporated into future prompts as demonstrations. This balances diversity and efficiency during the evolutionary search process.

## 3 OVERALL FRAMEWORK

Although LLM-based methods have demonstrated effectiveness in symbolic regression tasks, they still exhibit key limitations. As discussed earlier, existing approaches primarily rely on scalar evaluation metrics derived from raw data samples, overlooking the rich statistical characteristics of the dataset that can be extracted through program. These characteristics can provide valuable inductive biases, helping to constrain the search space and guide the discovery of correct symbolic expressions. This naturally raises the question: *Can statistical information enhance LLM-based SR?* In this section, we address this question through controlled analysis and then introduce our proposed framework, PROAUG.

### 3.1 CAN STATISTICAL INFORMATION ENHANCE LLM-BASED SR?

**Anaysis Setup.** To systematically examine the role of statistical information in LLM-based SR, we conduct a controlled case study on a representative instance, `II.6.15b_1_0`, from the LLM-

SRBench benchmark. The ground-truth expression for this case is: $A_{vec} = \frac{j \cdot m}{q \cdot \rho_{c_0}}$, which corresponds to a physical relationship involving the vector potential, current density, charge carrier density, elementary charge, and mass of a charge carrier. To eliminate confounding effects from prior knowledge, we anonymize all variables and remove the original physical background in the experimental conditions here. Specifically, inputs and outputs are replaced with $x_i$ and $y$ respectively. We then compare the following three settings:

- **Zero-shot**: The model receives no data samples or additional information.

- **Few-shot**: The model is provided with random 12 raw data samples without any additional background or statistical information.

- **Statistical Hint**: In addition to the 12 raw samples, the model is provided with a structured dictionary that summarizes dataset-level statistics. This dictionary includes (i) basic statistical measures for each variable and the output—such as mean, standard deviation, minimum, and maximum, and (ii) feature-level correlations, such as $R^2$ values between the logarithm of the output and simple transformations of the inputs (e.g., log, exp, sin, cos).

Prompts used here for all settings are provided in Appendix E. We use DeepSeek-V3.1 (Liu et al., 2024) as the base model within the LLM-SR framework, conducting a total of 100 evolutionary steps for each experiment.

**Results.** Figure 2 shows the best normalized mean squared error (NMSE) score trajectory under all settings. Both the zero-shot and few-shot settings perform poorly. The 12 randomly sampled data points provided in the few-shot condition are insufficient for the model to infer the true underlying expression. As a result, the NMSE decreases only slowly over the 100 evolutionary iterations. In contrast, the inclusion of statistical hints significantly accelerates convergence, ultimately yielding an NMSE on the order of $10^{-13}$. To understand the reason for this improvement, we look into the model's behavior under the statistical hint setting. We find that the high $R^2$ value between $\log y$ and $\log x$ provided leads the model to infer that $y$ is likely proportional to certain powers of the four input variables—suggesting a functional form of $y = x_0^a \cdot x_1^b \cdot x_2^c \cdot x_3^d$, where $a, b, c, d$ are parameters

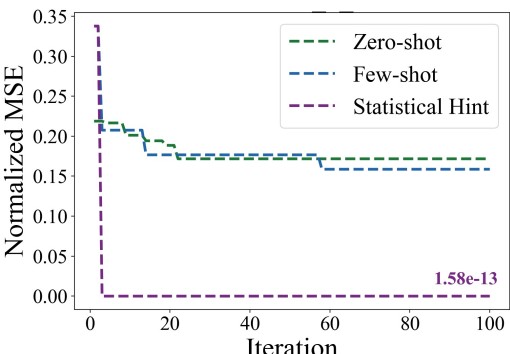

Figure 2: NMSE trajectories under three settings with varying amounts of background information.

can be optimized. This inferred form aligns perfectly with the ground-truth expression. These results demonstrate that even basic statistical information can provide substantial guidance in the LLM-SR process for the specific case. By constraining the search space and suggesting plausible functional forms, statistical cues can improve both the efficiency and the accuracy of symbolic regression. This mirrors how human scientists often practice exploratory data analysis before formulating hypotheses in the process of scientific discovery.

**Limitations.** Despite its utility, the design of the statistical context here has certain limitations. The correlation metrics are manually designed and confined to a small predefined set of transformations (e.g., exp, sin, cos). Consequently, complex nonlinear or multivariate interactions may remain undetected. While effective for highlighting simple algebraic relationships, this approach may fail with more intricate structures.

## 3.2 PROGRAMMATIC CONTEXT AUGMENTATION (PROAUG)

To fully leverage statistical context, it is essential to move beyond manually engineered heuristics. In the next section, we introduce **Programmatic Context Augmentation (PROAUG)**, in which LLMs—guided by physical knowledge—automatically generate dataset-analysis code. This enables

---

**Algorithm 1:** Pseudocode for PROAUG.

---

**Input:** Pretrained LLM $\mathcal{M}$, dataset $\mathcal{D}$, number of iterations $T$
**Output:** Best symbolic expression $f^{\star}$
**for** $t = 0, \ldots, T - 1$ **do**

    `// Programmatic context augmentation`
    Generate dataset-analysis prompt and sample analysis code
    Execute code on $\mathcal{D}$ to obtain informative signals
    `// Context construction`
    Construct enriched prompt using task instruction, problem specification, experience demonstrations,
     and extracted signals
    `// Hypothesis generation`
    Sample candidate equation from $\mathcal{M}$ given enriched prompt
    `// Optimization and evaluation`
    Optimize free parameters on $\mathcal{D}_{\text{tr-tr}}$
    Evaluate fitness score on $\mathcal{D}_{\text{tr-val}}$
    `// Experience management`
    Update population buffer with candidate and score
    Update best solution $f^{\star}$
**end**

---

the model to extract richer and more adaptive signals, preserving the benefits of inductive bias while broadening coverage.

The goal of PROAUG is to empower the LLM to actively interact with the dataset during evolutionary search by first proposing *data analysis programs*. These programs are executed to extract informative signals from the dataset, which are then incorporated into the prompt context. The enriched context prompt is subsequently used by the LLM to generate candidate symbolic expressions. In contrast, the original LLM-SR pipeline (see Section 2) bypasses this step: it directly prompts the LLM to generate hypotheses based only on task instructions, problem specifications, and past experience, without any dataset-level analysis.

Formally, PROAUG extends the LLM-SR pipeline by introducing an additional *dataset analysis phase* before hypothesis generation. The new pipeline proceeds as follows: 1. In the *data analysis phase*, The LLM is instructed with a structured prompt to generate Python code that extracts statistical or structural information from the dataset. 2. The generated code is executed, and the extracted signals (e.g., summary statistics, correlations, transformation relationships) are fed back as part of an enriched context prompt. 3. Using this enriched prompt, the LLM generates candidate equations within an evolutionary search framework similar as LLM-SR.

The structured prompt in the *data analysis phase* includes both a task description and a lightweight code template of basic analysis tools, which includes computations such as (i) descriptive statistics of each variable (mean, variance, range), (ii) linear correlations, and $R^2$ tests between transformed features (e.g., $\log, \exp, \sin, \cos$) or feature combinations and the target variable or the transformation of it.

A brief overview of PROAUG is shown in Algorithm 1, along with a pipeline illustration and an example of a PROAUG prompt, is provided in Figure 1 and Appendix E.

## 4 EXPERIMENTS

### 4.1 EXPERIMENTAL SETUP

**Datasets** Benchmarking LLMs in symbolic regression is challenging due to concerns that pretrained models may exploit memorized knowledge rather than demonstrating genuine reasoning and search capabilities. We evaluate PROAUG on LLM-SRBENCH (Shojaee et al., 2025), a recently proposed benchmark designed to mitigate this issue. LLM-SRBench consists of two primary subsets: **LSR-Transform** and **LSR-Synth**.

LSR-Transform includes 111 target functions adapted from Feynman equations (Udrescu & Tegmark, 2020), further modified through variable substitutions and symbolic rewrites to increase

Table 1: Comparison of different methods on LLM-SRBENCH, evaluated using normalized mean squared error (NMSE). The benchmark subsets contain 17 tasks from LSR-Transform, and 5, 3, 5, and 3 tasks from Chemistry, Biology, Physics, and Materials Science, respectively. Each entry reports the average NMSE across all tasks within the corresponding subset. The final column (Average) reports the mean NMSE over all LSR-Synth tasks (16 instances in total). Bold values indicate the best performance for each model, while underlined values denote the best overall performance across models.

| Models | LSR-Transform | LSR-Syn | | | | |
| --- | --- | --- | --- | --- | --- | --- |
| | | Chemistry | Biology | Physics | Material Science | Average |
| Qwen3-4B-Instruct-2507 (Yang et al., 2025) | | | | | | |
| LLM-SR | 0.258 | **7.55e-6** | 2.48 | **0.048** | **1.60e-6** | 0.480 |
| Statistical Hint | 0.225 | 0.032 | 0.095 | 0.22 | 0.012 | 0.099 |
| PROAUG | **0.145** | 4.73e-3 | **0.036** | 0.060 | 9.62e-6 | **0.027** |
| Qwen3-8B (Yang et al., 2025) | | | | | | |
| LLM-SR | 0.199 | 0.012 | 0.041 | 0.077 | 2.05e-4 | 0.036 |
| Statistical Hint | 0.279 | 4.11e-3 | 0.019 | **0.019** | 0.034 | 0.017 |
| PROAUG | **0.148** | **3.10e-3** | **1.08e-3** | 0.021 | **1.67e-7** | **7.75e-3** |
| Deepseek-V3.1 (Liu et al., 2024) | | | | | | |
| LLM-SR | 0.230 | 0.080 | **0.060** | 0.110 | 8.47e-7 | 0.071 |
| Statistical Hint | 0.188 | 0.022 | 1.112 | **0.034** | 0.333 | 0.288 |
| PROAUG | **0.067** | **1.02e-05** | 0.094 | 0.058 | **1.82e-9** | **0.036** |

structural complexity. Due to computational constraints, we randomly sample 15% of the tasks (17 instances) for evaluation. LSR-Synth consists of symbolic regression tasks that integrate canonical scientific terms with synthetically constructed expressions, spanning four domains—chemistry, physics, biology, and materials science. From this subset, we randomly select 16 tasks for testing. The complete list of tasks used in our experiments is provided in the Appendix B.

**Baselines**  We adopt LLM-SR (Shojaee et al., 2024) as our primary baseline and evaluate three backbone models: Qwen3-4B-Instruct-2507 (Yang et al., 2025), Qwen3-8B (Yang et al., 2025), and DeepSeek-V3.1 (Liu et al., 2024), covering a range of model scales. In addition to LLM-SR, we also compare against a hand-crafted statistical hint baseline, which incorporates both the problem specifications (physical meaning of variables) as well as the statistical hint. All methods are evolved for 150 iterations per task, with 2 samples generated per prompt. To reduce the effect of randomness, each experiments is repeated three times with different random seed, and we report the mean results.

**Evaluation Metrics**  Follow prior work, we primarily evaluate performance using the normalized mean squared error (NMSE). NMSE quantifies prediction error normalized by target variance: $\text{NMSE} = \frac{\sum_{i=1}^{N_{\text{test}}}(f(x_i)-y_i)^2}{\sum_{i=1}^{N_{\text{test}}}(y_i-\bar{y})^2}$, where $\bar{y}$ denotes the mean of the true values. By emphasizing large deviations, NMSE is well-suited for assessing numeric precision in symbolic regression.

## 4.2 RESULTS

As shown in Table 1, our method achieves lower NMSE on most datasets, consistently outperforming the LLM-SR baseline. For instance, in the LSR-Transform subset with DeepSeek-V3.1 as the backbone, our method attains an NMSE of 0.067 compared to 0.23 for LLM-SR, which is a 3x reduction in error. Moreover, we observe that the performance advantage of our method over LLM-SR becomes more pronounced as the capability of the underlying model improves. In particular, with DeepSeek models, our approach yields significant improvements across all tasks. We attribute this trend to the nature of programmatic context augmentation, which depends on the model's semantic understanding: weaker models are more likely to be overwhelmed by the extended context, while stronger models can effectively leverage it. Therefore, as base models continue to grow stronger

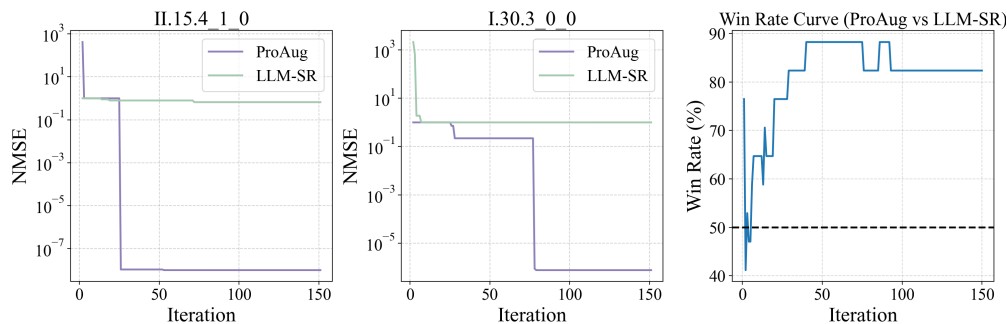

Figure 3: Analysis of convergence efficiency on LSR-Transform for DeepSeek-V3.1.

Table 2: SFT results using NMSE on LSR-Transform. best performance for each model in bold.

| LSR-Transform | Qwen3-4B-Instruct-2507 | Qwen3-8B |
|---|---|---|
| LLM-SR | 0.258 | 0.199 |
| LLM-SR-SFT | 0.110 | 0.131 |
| PROAUG | 0.145 | 0.148 |
| PROAUG-SFT | **0.107** | **0.106** |

in semantic understanding and context management, we expect our method to become even more effective in the near future.

### 4.3 FINETUNING

**Setting** Furthermore, we conduct supervised fine-tuning (SFT) for the Qwen3-4B-Instruct-2507 and Qwen3-8B models using samples drawn from the remaining portion of LSR-Transform. The full training set consists of 77 problems, and we employ DeepSeek-V3.1 for data distillation. The distillation data generation procedure is divided into two stages. In the first stage, the model is provided with the ground-truth expression and prompted to generate a data analysis program based on background knowledge and the correct answer. The generated program is then executed, and if execution fails, the generation process is retried up to ten attempts. In the second stage, we concatenate the results of the data analysis and prompt the model to produce the final expression together with its reasoning steps. The SFT model was trained on two NVIDIA A800 GPUs with a batch size of 16, a learning rate of 5e-6, and ZeRO Stage 2 for parallel acceleration.

We observe that the model occasionally generates non-executable code. To mitigate this issue, we augment the training data with additional examples where the data analysis code fails, but the model is still required to generate the correct expression. For comparison, we also distill outputs from DeepSeek-V3.1 for the LLM-SR method as an additional baseline. Finally, we omit SFT on LSR-Synth, as in this dataset subset, distinct problems share identical input prompts, leading to a one-to-many mapping issue that makes SFT ill-posed and counterproductive.

**Results** Table 2 reports the SFT results on LSR-Transform. We observe that supervised fine-tuning consistently improves performance for both LLM-SR and PROAUG, indicating that even standard fine-tuning provides clear benefits in this setting. For example, with Qwen3-4B-Instruct-2507, NMSE decreases from 0.258 to 0.110 for LLM-SR after SFT. Similarly, PROAUG also benefits from fine-tuning, with NMSE reduced from 0.145 to 0.107. Importantly, PROAUG-SFT achieves the lowest errors overall across both backbones, highlighting that our method remains effective and complementary to supervised fine-tuning.

### 4.4 METHOD BEHAVIOR ANALYSIS

#### 4.4.1 CONVERGENCE EFFICIENCY

We evaluated the convergence efficiency of PROAUG relative to LLM-SR on the LSR-Transform subset from the DeepSeek-V3.1 model, with results shown in Figure 3. The two plots on the left

provide representative examples, illustrating that our method achieves a more rapid reduction in NMSE on train-val set during the iterative search process and ultimately converges to a lower error. The plot on the right displays the win rate of PROAUG over LLM-SR across 17 LSR-Transform test cases as a function of the number of iterations. We observe that PROAUG starts with a high win rate (above 50%) early in the process, indicating fast convergence of scores in the initial stages. As the iterations progress, the win rate continues to increase, eventually exceeding 80%. These results demonstrate that PROAUG maintains superior performance while also converges efficiently.

### 4.4.2 VARIANCE ANALYSIS

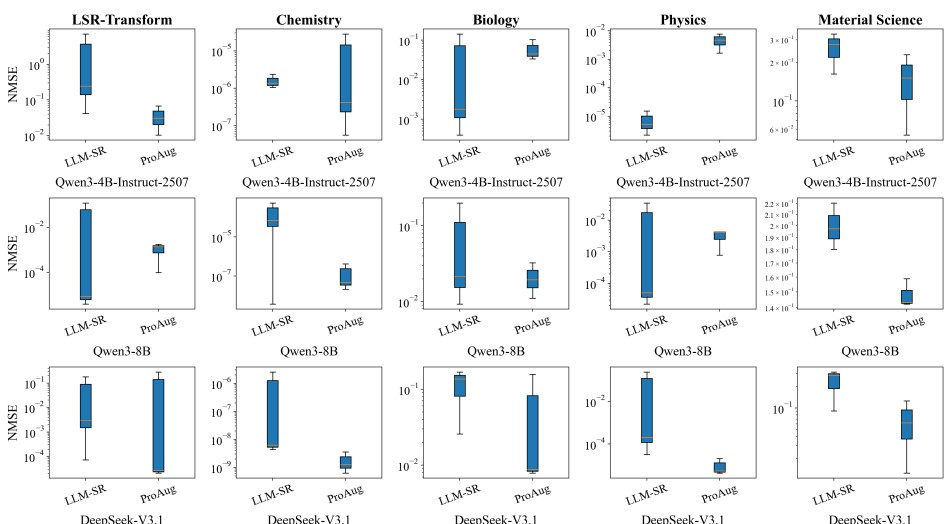

Figure 4: Boxplot comparison of tri-run variance for LLM-SR vs. PROAUG.

To rigorously evaluate the stability and reproducibility of LLM-SR and PROAUG, we conducted multiple independent runs under controlled hyperparameters and random seeds. Figure 4 presents boxplots summarizing the distribution of results, including the median and interquartile range (IQR) for each method.

Overall, PROAUG exhibits substantially lower variance across runs. LLM-SR shows high sensitivity to randomness, with outcomes occasionally differing by several orders of magnitude. For example, on the Chemistry task with Qwen3-8B, the NMSE varied from $10^{-4}$ to $10^{-9}$ across runs. Such findings highlight a critical gap in current LLM-based SR research, where multi-run evaluations are rarely reported. We therefore recommend that future studies adopt repeated trials to mitigate randomness and enhance reliability.

## 5 RELATED WORK

**Symbolic Regression.** Symbolic regression (SR) is core to scientific discovery. Interest began in the 1970s (Gerwin, 1974; Langley, 1977) and SR has recently become prominent in AI-for-science (Makke & Chawla, 2024; Merler et al., 2024; Romera-Paredes et al., 2024). SR methods follow three algorithmic themes:

*Search based methods.* SR is a challenging combinatorial optimization task. These methods search the space of possible equations while minimizing an objective function, relying on custom heuristics and parallelization to efficiently hypothesize and evaluate candidates (Cranmer, 2023; Petersen et al., 2019; Stephens, 2024; Udrescu & Tegmark, 2020). Notably, PySR (Cranmer, 2023) presents a multi-population evolutionary algorithm that has found practical utility in cosmology (Davis & Jin, 2023), international economics (Verstyuk & Douglas, 2022), and climate modeling (Grundner et al., 2024). While such approaches work well out-of-the-box, they are difficult to steer towards specific class of equations. PROAUG extends such approaches by providing two flexible steering mechanisms:

scientists can either manually steer search through natural language priors or such priors can be automatically extracted using dataset statistics.

*Learning based methods.* Another approach uses neural networks to accelerate SR (Tenachi et al., 2023; Biggio et al., 2021; Landajuela et al., 2022). These methods train networks on experimental data to either directly induce target expressions (Merler et al., 2024; Romera-Paredes et al., 2024; Meyerson et al., 2024; Biggio et al., 2021) or to accelerate the search for target expressions (Tenachi et al., 2023; Romera-Paredes et al., 2024; Zhang et al., 2025; Shah et al., 2020). However, neural networks require retraining when datasets change significantly, necessitating scientists to monitor data distribution shifts and repeatedly retrain models – both computationally expensive tasks. PROAUG follows prior work in primarily being a data-driven SR algorithm, but avoids the computational overheads by replacing learned neural representations with programmatically extracted insights, greatly increasing practical utility. In particular, compared with RAG-SR (Zhang et al., 2025), PROAUG focuses on improving LLM-based SR methods that leverage the physical reasoning and world knowledge capabilities of large language models as agents (e.g., LLM-SR, LaSR). In contrast, RAG-SR employs transformer-based neural networks without LLM-scale world knowledge or physical reasoning components, representing a fundamentally different methodological paradigm. Our methodology of leveraging programmatic interaction with the dataset to extract complex statistical insights for enhancing reasoning and search is also fundamentally different from RAG-SR. RAG-SR reconciles with our motivation that scalar feedback can be limited, while complementarily work on different lines of methods.

*LLM based methods.* Recent works leverage large language models for symbolic regression (Romera-Paredes et al., 2024; Novikov et al., 2025; Shojaee et al., 2024; Grayeli et al., 2024; Ma et al., 2024). These methods rely on LLMs' world knowledge to propose and mutate programs conditioned on natural language instructions and execution feedback. Specifically, Funsearch (Romera-Paredes et al., 2024) and AlphaEvolve (Novikov et al., 2025) demonstrate that LLMs as mutation operators within evolutionary algorithms can discover novel heuristics for long-standing problems in mathematics, hardware design, and algorithms. Many exciting works augment the evolutionary process with sketching (Shojaee et al., 2024), library learning (Grayeli et al., 2024), and tool use (Ma et al., 2024). PROAUG leverages pretrained LLMs similarly but treats the scientific dataset as a first-class citizen in the optimization process rather than merely a scoring mechanism. Consecutively, PROAUG's primary contribution is *complementary* to that of these methods, and it is technically possible to employ programmatic context augmentation to enhance other SR algorithms.

**Scientific Discovery with Foundation Models.** Modern scientific breakthroughs increasingly demand both mastery of individual fields and trans-disciplinary insights (Gottweis et al., 2025; Jinek et al., 2012). As such, many LLM frameworks have emerged that assist scientists in various stages of the scientific process – from hypothesis falsification (Huang et al., 2025) and idea generation (Radensky et al., 2025) to data-driven discovery (Majumder et al., 2024), heuristic design (Romera-Paredes et al., 2024; Novikov et al., 2025), and multi-purpose systems spanning hypothesis generation to experiment design (Gottweis et al., 2025; Lu et al., 2024). While these systems primarily interface with knowledge sources and scientists through natural language, PROAUG (like (Romera-Paredes et al., 2024; Novikov et al., 2025)) interfaces with these sources through code – leveraging the precise semantics and deterministic execution of code to rigorously analyze scientific data, in addition to using textual reasoning.

## 6 CONCLUSION

In this work, we address a key limitation of existing LLM-based symbolic regression methods: their reliance on simplistic scalar feedback such as mean squared error. We propose PROAUG, a novel framework that enriches the model's interaction with data by assigning the LLM a dual role—generating both candidate equations and data-analysis code. This design fosters a more active, human-like discovery process that leverages rich statistical signals and variable relationships. Experimental results on the challenging LLM-SRBENCH benchmark highlight the clear advantages of our approach. We believe this work represents a step toward more intelligent and interactive AI systems with stronger capabilities for scientific reasoning and discovery.

**Limitations and future work**   Despite its promising results, our approach is not without limitations. The current framework, while leveraging richer statistical feedback than MSE-based methods, still relies on a predefined set of data analysis operations. This may restrict its ability to discover equations that depend on highly complex or non-standard data relationships not captured by our initial code-generation template. Future work could explore enabling the LLM to propose entirely novel analytical procedures.

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

## A    USE OF LLMs

In the course of this work, we made limited use of LLMs as auxiliary tools. Specifically, we used an LLM to assist in generating small utility Python scripts for tasks such as data preprocessing, formatting, and visualization. These scripts were only supportive in nature and did not contribute to the conceptual development, design of experiments, or the core scientific results of the paper. All key research ideas, experimental design, and writing of the main paper were carried out independently by the authors. The LLM was not used to generate or edit the scientific content of the manuscript, nor was it involved in producing research hypotheses or interpretations. The authors take full responsibility for the correctness, originality, and integrity of all content in the paper.

## B    PROBLEM DETAILS

Details of the problems we used in our experiments are shown in Table 3.

Table 3: Test data we use from LLM-SRBench.

| Discipline | Name |
|---|---|
| Chemistry | CRK13, CRK12, CRK35, CRK31, CRK8 |
| Biology | BPG16, BPG8, BPG9 |
| Physics | PO12, PO37, PO40, PO24, PO8 |
| Material Science | MatSci9, MatSci18, MatSci8 |
| LSR-Transform | I.29.16_1_0,  II.6.15b_1_0,  II.11.27_1_0,  I.34.1_2_0,  I.30.3_0_0, II.24.17_1_1,  III.15.27_2_0,  I.12.2_3_0,  III.10.19_2_1,  I.37.4_0_1, II.11.17_4_0,  II.6.15b_3_0,  II.15.4_1_0,  II.13.23_1_0,  II.34.29b_3_0, I.11.19_2_0, I.32.5_1_1 |

Table 4: Results of $Acc_{0.1}$.

| Models | LSR-Transform | LSR-Synth | | | | |
|--------|---------------|-----------|---------|---------|------------------|---------|
| | | Chemistry | Biology | Physics | Material Science | Average |
| *Qwen3-4B-Instruct-2507* (Yang et al., 2025) | | | | | | |
| LLM-SR | 59.9±4.2 | 72.5±0.8 | 44.2±26.3 | 82.6±15.5 | 98.0±1.3 | 75.1±7.3 |
| PROAUG | 65.0±3.2 | 71.3±3.8 | 36.4±1.8 | 58.0±10.8 | 99.2±0.3 | 65.8±4.6 |
| *Qwen3-8B* (Yang et al., 2025) | | | | | | |
| LLM-SR | 68.0±1.4 | 72.1±2.3 | 62.2±23.7 | 63.8±8.9 | 92.9±7.1 | 71.6±3.6 |
| PROAUG | 57.5±0.9 | 73.2±14.4 | 39.0±12.2 | 65.3±11.3 | 98.7±1.5 | 69.1±4.5 |
| *Deepseek-V3.1* (Liu et al., 2024) | | | | | | |
| LLM-SR | 70.1±4.1 | 65.2±10.5 | 68.0±13.7 | 56.3±5.5 | 99.5±0.7 | 69.4±7.0 |
| PROAUG | 78.3±5.2 | 69.5±1.7 | 68.8±12.8 | 70.4±8.5 | 99.9±0.1 | 75.3±5.1 |

## C   OTHER EXPERIMENT RESULTS

We provide results of Accuracy under error tolerance ($ACC_\tau$) which measures numeric precision. Specifically $ACC_\tau$ is defined as the proportion of test points with error below a threshold $\tau$: $ACC_\tau = \frac{1}{N_{\text{test}}} \sum_{i=1}^{N_{\text{test}}} \mathbf{1}\left( \left| \frac{f(x_i) - y_i}{y_i} \right| \leq \tau \right)$, where $\mathbf{1}(\cdot)$ is the indicator function. $ACC_\tau$ captures "good enough" performance under bounded error, aligning with real-world tolerance requirements. We show results in Table 4.

# D EQUATION EXAMPLES

We present several representative equations discovered by PROAUG and demonstrate how LLM-generated statistical information enhances algorithmic performance.

## D.1 EQUATION EXAMPLES

The following examples illustrate equations discovered by PROAUG.

---

**Example 1: I.32.5_1_1**

```
Problem background: radiation from accelerating charges in classical
↪  electrodynamics.
Ground truth:  sqrt(6)*sqrt(pi)*sqrt(Pwr*c**3*epsilon)/q
NMSE: 3.15e-14
Symbol meaning: [Target: 'the acceleration of the charged particle',
↪  Feature: 'the power of an electromagnetic wave', 'the charge of a
↪  particle', 'the electric constant or permittivity of free space',
↪  'the speed of light']

``` Discovered equation:
def equation(Pwr: np.ndarray, q: np.ndarray, epsilon: np.ndarray, c:
↪  np.ndarray, params: np.ndarray) -> np.ndarray:
    term1 = np.power(Pwr, params[0])
    term2 = np.power(q, params[1])
    term3 = np.power(epsilon, params[2])
    term4 = np.power(c, params[3])
    output = params[4] * term1 * term2 * term3 * term4 + params[5]
    return output
```

---

**Example 2: I.37.4_0_1**

```
Problem background: interference and intensity relations for two
↪  coherent wave sources.
Ground truth: 2*I2*cos(delta)**2 + I2 + Int +
↪  2*sqrt(I2*(I2*cos(delta)**2 + I2 + Int))*cos(delta)
NMSE: 2.68e-14
Symbol meaning: [Target: 'the intensity of the first wave source',
↪  Feature: 'the resultant intensity of two wave sources', 'the
↪  intensity of the second wave source', 'the phase difference
↪  between the two wave sources']

``` Discovered equation:
def equation(Int: np.ndarray, I2: np.ndarray, delta: np.ndarray,
↪  params: np.ndarray) -> np.ndarray:
    cos_term = np.cos(delta + params[2])
    sqrt_I2 = np.sqrt(np.maximum(I2, 1e-8))
    a = 1.0
    b = params[1] * cos_term * sqrt_I2
    c = params[0] * I2 - Int
    discriminant = b**2 - 4*a*c
    valid_mask = discriminant >= 0
    sqrt_I1 = np.zeros_like(Int)
    sqrt_I1[valid_mask] = (-b[valid_mask] +
    ↪  np.sqrt(discriminant[valid_mask])) / (2*a)
    I1_result = np.maximum(sqrt_I1**2 + params[3] + params[4] * I2 *
    ↪  cos_term, 0)
    return I1_result
```

---

> **Example 3: III.15.27_2_0**
>
> ```
> Problem background: quantized wave modes in periodic physical
> ↪  systems.
> Ground truth: 2*pi*alpha/(k*n)
> NMSE: 3.54e-15
> Symbol meaning: [Target: 'the diameter or a characteristic length',
> ↪  Feature: 'the spring constant or a proportionality constant', 'a
> ↪  material-dependent constant or a dimensionless parameter', 'the
> ↪  number of turns or a dimensionless quantity']
> ``` Discovered equation:
> def equation(k: np.ndarray, alpha: np.ndarray, n: np.ndarray, params:
> ↪  np.ndarray) -> np.ndarray:
>     output = params[0] * alpha / (k * n)
>     return output
> ```
> ```

These examples demonstrate PROAUG's ability to discover accurate equations across diverse problems with low NMSE that closely match the ground truth.

## D.2 LLM-GENERATED STATISTICAL INFORMATION

We provide a concrete example demonstrating how GPT-generated statistical information improves performance on MatSci9 from the Material Science category in LSR-Synth. The problem is about finding the equation that represents Stress (sigma), given data on Strain (epsilon), and Temperature (T).

PROAUG generates code to extract statistical information including correlation and regression r2 values for LLM-proposed features including `epsilon * T, epsilon ** 3, epsilon ** 2, epsilon - T`, etc.

Given statistical context, the LLM reasoned that

> "From the dataset and the extracted information, we can observe several important trends and correlations. The output 'sigma' (Stress) shows a strong correlation with both 'epsilon' (Strain) and 'T' (Temperature), with a correlation coefficient of 0.693 for both. The $R^2$ values also support this, indicating that both variables contribute significantly to predicting Stress. The strongest correlation with 'sigma' is observed with 'epsilon_squared', followed by 'epsilon_cubed', and 'epsilon_minus_T'. Additionally, the $R^2$ values for 'sigma_vs_epsilon_squared' and 'sigma_vs_T_squared' suggest that non-linear relationships may be more appropriate than linear ones. Given the high correlation with 'epsilon_squared' and the fact that Stress is often related to the square of Strain in physical models (such as in Hooke's Law or stress-strain relationships), we can infer that the function might be a quadratic or cubic function of Strain, possibly combined with Temperature.",

which leads to the discovered equation 'output = params[0] * epsilon + params[1] * T + params[2] * epsilon**3 + params[3] * epsilon**3 * T' after several search iterations.

The discovered equation, with a NMSE of 8.029e-14, closely matches the ground truth, which is "4.255830478818137*epsilon**3 - 0.39029568781474044*(T - 308.8646781296164) + epsilon**3*8.63802815432207*(T - 308.8646781296164)", demonstrating how statistical context guides the LLM toward physically meaningful functional forms.

# E  PROMPT DESIGN

## E.1  PROAUG PROMPT DESIGN

Aside from the statistical information augmentation component, we adopt the same prompt structure as the original LLM-SR/LLM-SRBench framework. The prompt template below illustrates the structure of the prompt used in PROAUG.

---

**PROAUG Prompt Template**

You are a helpful assistant tasked with discovering mathematical function structures for scientific data science tasks.  Complete the 'equation' function below, considering the dataset information.
"""

Find the mathematical function skeleton that represents $OUTPUT_VAR_DESC, given data on $INPUT_VAR_DESC[0], ..., $INPUT_VAR_DESC[N].
"""

```python
import numpy as np

#Initialize parameters
MAX_NPARAMS = 10
params = [1.0]*MAX_NPARAMS

def equation_v0($INPUT_VAR[0], ..., $INPUT_VAR[N], params):
    """
    Args:
        $INPUT_VAR[0]: A numpy array representing observations of
        ↪  {$INPUT_VAR_DESC[0]}.
        ...
        $INPUT_VAR[N]: A numpy array representing observations of
        ↪  {$INPUT_VAR_DESC[N]}.
        params: Array of numeric constants or parameters to be
        ↪  optimized

    Return:
        A numpy array representing {$OUTPUT_VAR_DESC} as the result
        ↪  of applying the mathematical function to the inputs.
    """
    # Equation example 1 logic as function body
    ...

def equation_v1($INPUT_VAR[0], ..., $INPUT_VAR[N], params):
    """Improved version of `equation_v0`."""
```

You are to write Python code to extract and analyze relevant information from the dataset, with the goal of uncovering the underlying scientific function structure. The code will be executed in a sandbox environment that has direct access to the data.  The output will be returned to assist your reasoning and support you in deriving the final answer.
You can do, but not limited to, investigating common statistics, use physical reasoning to propose candidate engineered features (inverse, cos/sin, log, exp, ratio, products, powers, etc) as well as compositions of these terms, and use tools including but not limited to correlation analysis, log-log fit, log-linear fit, multivariate linear fits, periodicity tests, and additive models in log space, etc, to identify whether the system follows additive, multiplicative, exponential, or power-law relationships.

```python
def extract_dataset_info($OUTPUT_VAR, $INPUT_VAR[0], ...,
↪  $INPUT_VAR[N]):
    """
    Args:
        $OUTPUT_VAR[0]: A numpy array representing observations of
        ↪  {$OUTPUT_VAR_DESC}.
```

---

```
        $INPUT_VAR[0]: A numpy array representing observations of
        ↪   {$INPUT_VAR_DESC}.
        ...
        $INPUT_VAR[N]: A numpy array representing observations of
        ↪   {$INPUT_VAR_DESC[N]}.

    Return:
        A dict summarizing the extracted dataset information.
    """
    import numpy as np
    from scipy.stats import linregress
    def r2(x, y):
        slope, intercept, r_value, _, _ = linregress(x, y)
        return float(np.round(r_value ** 2, 3))

    # Basic statistics
    """ mean / std of certain variables """
    info['stats'] = {}

    # Feature transform
    """ use physical reasoning to propose a variety of candidate
    ↪   engineered features as well as compositions of these terms
    ↪   for [$OUTPUT_VAR, $INPUT_VAR[0], ..., $INPUT_VAR[N]]"""
    features = {}

    # Linear correlation with $OUTPUT_VAR
    info['correlations_with_$OUTPUT_VAR'] = {}

    # R2 values from regression fits
    """ regression fits between $OUTPUT_VAR or transforms of
    ↪   $OUTPUT_VAR and [$INPUT_VAR[0], ..., $INPUT_VAR[N]] and
    ↪   features of [$INPUT_VAR[0], ..., $INPUT_VAR[N]] """

    info['regression_r2'] = {}
    return info
info = {'stats': {}, 'correlations_with_Int_0': {}, 'regression_r2':
↪   {}}
```

Given this template, the LLM first generates the statistical analysis code function `extract_dataset_info`, which is executed in a sandbox environment with direct access to the data. The output is then returned to the LLM to generate the final equation.

**Computational cost:** Compared to LLM-SR, The main additional overhead of PROAUG is that requires two LLM calls per iteration (one for statistical code generation, one for equation generation) versus one call for original LLM-SR. However, this can be viewed as a single tool-use process, where the LLM generates analysis tools (statistical code), obtains tool-call results (extracted statistics), and performs the final action (equation generation). As LLM inference techniques continue to advance and become more efficient, we expect this overhead to diminish further.

For comparison, the LLM-SR prompt template is shown as follows:

**LLM-SR Prompt Template**

You are a helpful assistant tasked with discovering mathematical function structures for scientific data science tasks. Complete the 'equation' function below, considering the physical meaning and relationships of inputs.
'''''

Find the mathematical function skeleton that represents $OUTPUT_VAR_DESC, given data on $INPUT_VAR_DESC[0], ..., $INPUT_VAR_DESC[N].
'''''

```
import numpy as np
```

```
#Initialize parameters
MAX_NPARAMS = 10
params = [1.0]*MAX_NPARAMS

def equation_v0($INPUT_VAR[0], ..., $INPUT_VAR[N], params):
    """
    Args:
        $INPUT_VAR[0]: A numpy array representing observations of
        ↪  {$INPUT_VAR_DESC[0]}.
        ...
        $INPUT_VAR[N]: A numpy array representing observations of
        ↪  {$INPUT_VAR_DESC[N]}.
        params: Array of numeric constants or parameters to be
        ↪  optimized

    Return:
        A numpy array representing {$OUTPUT_VAR_DESC} as the result
        ↪  of applying the mathematical function to the inputs.
    """
    # Equation example 1 logic as function body
    ...

def equation_v1($INPUT_VAR[0], ..., $INPUT_VAR[N], params):
    """Improved version of `equation_v0`."""
```

As shown, PROAUG shares the same prompt structure as LLM-SR, aside from the statistical information augmentation component. Both LLM-SR and PROAUG use a linear regression model as the seed hypothesis at the beginning of the search process. For both LLM-SR and PROAUG, the prompts for different problem instances are generated by replacing the input and output variable names $OUTPUT_VAR, $INPUT_VAR[0], ..., $INPUT_VAR[N], and short descriptions $OUTPUT_VAR_DESC, $INPUT_VAR_DESC[0], ..., $INPUT_VAR_DESC[N] in the template with instance-specific values. Finally, we provide a concrete example using a linear regression equation for instance II.6.15b_1_0 to illustrate what these variable names and descriptions look like.

**Equation example for `II.6.15b_1_0`**

```
"""
Find the mathematical function skeleton that represents the dipole moment, given data on the
electric field, the electric constant or permittivity of the medium, the angle between the dipole
axis and the position vector, and the distance from the dipole to the point where the electric
field is being measured.
"""

import numpy as np

#Initialize parameters
MAX_NPARAMS = 10
params = [1.0]*MAX_NPARAMS

def equation_v0(Ef, epsilon, theta, r, params):
    """
    Args:
        Ef: A numpy array representing observations of the electric
        ↪  field.
        epsilon: A numpy array representing observations of the
        ↪  electric constant or permittivity of the medium.
        theta: A numpy array representing observations of the angle
        ↪  between the dipole axis and the position vector.
```

```
        r: A numpy array representing observations of the distance
        ↪  from the dipole to the point where the electric field is
        ↪  being measured.
        params: Array of numeric constants or parameters to be
        ↪  optimized

    Return:
        A numpy array representing the dipole moment as the result of
        ↪  applying the mathematical function to the inputs.
    """
    output = params[0] * Ef + params[1] * epsilon + params[2] * theta
    ↪  + params[3] * r + params[4]
    return output
```

## E.2 PROMPTS USED IN SECTION 3.1

We present the Zero-shot, Few-shot, and Statistical hint prompts used in Section 3.1 for instance `II.6.15b_1_0` as follows.

---

**Zero-Shot Prompt**

You are a helpful assistant tasked with discovering mathematical function structures for scientific data science tasks. Complete the 'equation' function below, considering the dataset information.

```
"""
### Your Task:
FIRST, provide **BRIEF** reasoning inside a <thought>...</thought>
↪   block.

THEN, AFTER THE </thought> BLOCK, output your evolved mathematical
↪   function skeleton in Python. The function should begin with a
↪   'def' line and follow standard Python formatting.

Note: DO NOT use more than 10 params
"""

import numpy as np

#Initialize parameters
MAX_NPARAMS = 10
params = [1.0]*MAX_NPARAMS

def equation_v0(x_0: np.ndarray, x_1: np.ndarray, x_2: np.ndarray,
↪   x_3: np.ndarray, params: np.ndarray) -> np.ndarray:
    """ Mathematical function

    Args:
        x_0: A numpy array.
        x_1: A numpy array.
        x_2: A numpy array.
        x_3: A numpy array.

        params: Array of numeric constants or parameters to be
        ↪   optimized

    Return:
        A numpy array as the result of applying the mathematical
        ↪   function to the inputs.
    """
    output = params[0] * x_0 + params[1] * x_1 + params[2] * x_2 +
    ↪   params[3] * x_3 + params[4]
    return output

def equation_v1(x_0: np.ndarray, x_1: np.ndarray, x_2: np.ndarray,
↪   x_3: np.ndarray, params: np.ndarray) -> np.ndarray:
    """Improved version of `equation_v0`."""
```

---

**Few-shot Prompt**

You are a helpful assistant tasked with discovering mathematical function structures for scientific data science tasks. Complete the 'equation' function below, considering the dataset information.

```
"""
Find the mathematical function skeleton that represents y, given data
↪  on x_0, x_1, x_2, x_3.

### 12 Random Samples (X, Y) (Sorted by Y from small to large):
X[0] = [-32.193, 4.357, 4.774, 1.672], Y[0] = 2.588
X[1] = [-8.799, 2.543, 2.522, 2.127], Y[1] = 2.918
X[2] = [-45.200, 4.476, 3.700, 2.379], Y[2] = 6.494
X[3] = [-64.941, 3.223, 3.793, 1.486], Y[3] = 7.892
X[4] = [-46.401, 3.874, 4.408, 4.287], Y[4] = 11.650
X[5] = [-76.242, 2.469, 2.298, 1.248], Y[5] = 16.766
X[6] = [-84.225, 4.124, 1.593, 1.798], Y[6] = 23.048
X[7] = [-72.211, 3.124, 3.184, 3.350], Y[7] = 24.319
X[8] = [-43.300, 1.639, 4.213, 4.297], Y[8] = 26.954
X[9] = [-41.009, 1.516, 3.111, 3.784], Y[9] = 32.907
X[10] = [-83.370, 3.689, 2.829, 4.502], Y[10] = 35.962
X[11] = [-43.438, 1.281, 2.523, 3.718], Y[11] = 49.963

### Your Task:
FIRST, provide **BRIEF** reasoning inside a <thought>...</thought>
↪  block.

THEN, AFTER THE </thought> BLOCK, output your evolved mathematical
↪  function skeleton in Python. The function should begin with a
↪  'def' line and follow standard Python formatting.

Note: DO NOT use more than 10 params
"""

import numpy as np

#Initialize parameters
MAX_NPARAMS = 10
params = [1.0]*MAX_NPARAMS

def equation_v0(x_0: np.ndarray, x_1: np.ndarray, x_2: np.ndarray,
↪  x_3: np.ndarray, params: np.ndarray) -> np.ndarray:
    """ Mathematical function

    Args:
        x_0: A numpy array.
        x_1: A numpy array.
        x_2: A numpy array.
        x_3: A numpy array.

        params: Array of numeric constants or parameters to be
        ↪  optimized

    Return:
        A numpy array as the result of applying the mathematical
        ↪  function to the inputs.
    """
    output = params[0] * x_0 + params[1] * x_1 + params[2] * x_2 +
    ↪  params[3] * x_3 + params[4]
    return output

def equation_v1(x_0: np.ndarray, x_1: np.ndarray, x_2: np.ndarray,
↪  x_3: np.ndarray, params: np.ndarray) -> np.ndarray:
    """"Improved version of `equation_v0`."""
```

**Statistical Hint Prompt**

You are a helpful assistant tasked with discovering mathematical function structures for scientific data science tasks. Complete the 'equation' function below, considering the dataset information.

```
"""
Find the mathematical function skeleton that represents y, given data
↪  on x_0, x_1, x_2, x_3.
The information of (X, Y) dataset including random sample points,
↪  smoothness estimation and simple basis fit scores are as follows:
### 12 Random Samples (X, Y) (Sorted by Y from small to large):
X[0] = [-32.193, 4.357, 4.774, 1.672], Y[0] = 2.588
X[1] = [-8.799, 2.543, 2.522, 2.127], Y[1] = 2.918
X[2] = [-45.200, 4.476, 3.700, 2.379], Y[2] = 6.494
X[3] = [-64.941, 3.223, 3.793, 1.486], Y[3] = 7.892
X[4] = [-46.401, 3.874, 4.408, 4.287], Y[4] = 11.650
X[5] = [-76.242, 2.469, 2.298, 1.248], Y[5] = 16.766
X[6] = [-84.225, 4.124, 1.593, 1.798], Y[6] = 23.048
X[7] = [-72.211, 3.124, 3.184, 3.350], Y[7] = 24.319
X[8] = [-43.300, 1.639, 4.213, 4.297], Y[8] = 26.954
X[9] = [-41.009, 1.516, 3.111, 3.784], Y[9] = 32.907
X[10] = [-83.370, 3.689, 2.829, 4.502], Y[10] = 35.962
X[11] = [-43.438, 1.281, 2.523, 3.718], Y[11] = 49.963
Statistics: {'mean_Y': 21.331, 'std_Y': 24.473, 'min_Y': 0.03,
↪  'max_Y': 335.661, 'mean_X_0': -44.025, 'std_X_0': 25.099,
↪  'min_X_0': -87.583, 'max_X_0': -0.436, 'mean_X_1': 3.011,
↪  'std_X_1': 1.151, 'min_X_1': 1.0, 'max_X_1': 5.0, 'mean_X_2':
↪  2.995, 'std_X_2': 1.156, 'min_X_2': 1.0, 'max_X_2': 5.0,
↪  'mean_X_3': 3.007, 'std_X_3': 1.156, 'min_X_3': 1.0, 'max_X_3':
↪  5.0, 'r2_Y_X_0': 0.245, 'r2_Y_X_1': 0.151, 'r2_Y_X_2': 0.15,
↪  'r2_Y_X_3': 0.111, 'r2_log(Y)_log(X_0)': 0.595,
↪  'r2_log(Y)_log(X_1)': 0.133, 'r2_log(Y)_log(X_2)': 0.133,
↪  'r2_log(Y)_log(X_3)': 0.134, 'r2_log(Y)_log(sin(x_0))': 0.0,
↪  'r2_log(Y)_log(cos(x_0))': 0.0, 'r2_log(Y)_log(exp(x_0))': 0.433,
↪  'r2_log(Y)_log(sqrt(x_0))': 0.595, 'r2_log(Y)_log(sin(x_1))':
↪  0.003, 'r2_log(Y)_log(cos(x_1))': 0.001,
↪  'r2_log(Y)_log(exp(x_1))': 0.128, 'r2_log(Y)_log(sqrt(x_1))':
↪  0.133, 'r2_log(Y)_log(sin(x_2))': 0.004,
↪  'r2_log(Y)_log(cos(x_2))': 0.001, 'r2_log(Y)_log(exp(x_2))':
↪  0.128, 'r2_log(Y)_log(sqrt(x_2))': 0.133,
↪  'r2_log(Y)_log(sin(x_3))': 0.003, 'r2_log(Y)_log(cos(x_3))':
↪  0.001, 'r2_log(Y)_log(exp(x_3))': 0.129,
↪  'r2_log(Y)_log(sqrt(x_3))': 0.134}

### Your Task:
Based on the statistics above, analyze and evolve the mathematical
↪  function skeleton.

IMPORTANT:
FIRST, provide **BRIEF** reasoning inside a <thought>...</thought>
↪  block about how you interpret the data and how it guides your
↪  function structure decisions.

THEN, AFTER THE </thought> BLOCK, output your evolved mathematical
↪  function skeleton in Python. The function should begin with a
↪  'def' line and follow standard Python formatting.
```

```
Note: DO NOT use more than 10 params
"""

import numpy as np

#Initialize parameters
MAX_NPARAMS = 10
params = [1.0]*MAX_NPARAMS

def equation_v0(x_0: np.ndarray, x_1: np.ndarray, x_2: np.ndarray,
↪  x_3: np.ndarray, params: np.ndarray) -> np.ndarray:
    """ Mathematical function

    Args:
        x_0: A numpy array.
        x_1: A numpy array.
        x_2: A numpy array.
        x_3: A numpy array.

        params: Array of numeric constants or parameters to be
        ↪  optimized

    Return:
        A numpy array as the result of applying the mathematical
        ↪  function to the inputs.
    """
    output = params[0] * x_0 + params[1] * x_1 + params[2] * x_2 +
    ↪  params[3] * x_3 + params[4]
    return output

def equation_v1(x_0: np.ndarray, x_1: np.ndarray, x_2: np.ndarray,
↪  x_3: np.ndarray, params: np.ndarray) -> np.ndarray:
    """"Improved version of `equation_v0`."""
```