# OpenReview forum: "Programmatic Context Augmentation for LLM-based Symbolic Regression"
_ICLR.cc/2026/Conference — Submitted to ICLR 2026_

### Official Review · Reviewer_B4jH · 2025-10-28

[review text omitted: it was posted to a different submission]

---

> ### Author Response · Authors · 2025-11-13
> **Mismatched Review**
>
> We have observed that the "Official Review" submitted by Reviewer B4jH does not match our paper. The comments within this review clearly pertain to a different submission but were erroneously attributed to ours.
>
> We would greatly appreciate your urgent assistance in investigating this matter and ensuring the review records for our submission are corrected.
>
> Thank you very much for your time and help.

---

> ### Author Response · Authors · 2025-11-28
>
> We sincerely thank the reviewer for the constructive feedback. We address your concerns as follows:
>
>
> **Regarding Weaknesses 1 & 4, Questions 1 & 5 (Templates Design and Underlying Mechanisms):**
>
> We respectfully clarify that our template design does not limit the model's ability to capture complex nonlinear and multivariate relationships. While we provide fixed templates and statistical tools (such as R² and correlation), the LLM autonomously proposes meaningful nonlinear and multivariate features based on the problem's physical semantics. The template serves as a framework, not a constraint.
>
> One key goal of ProAug is to provide an accessible mechanism for scientists to inject domain knowledge into equation discovery. A fixed, verifiable, and well-understood augmentation template is often more valuable than having an adaptive but uncontrollable one. It enables scientists to iteratively refine the possible augmentations to suit their domain, facilitating collaborative human-AI discovery. Our current operations were designed to enable general-purpose augmentations across scientific domains such as physics and chemistry.
>
> To demonstrate the mechanism by which discovered statistical features guide equation discovery, consider the following example:
>
> The problem is about finding the equation that represents Stress (sigma), given data on Strain (epsilon), and Temperature (T).
>
> ProAug generates code to extract statistical information including correlation and regression r2 values for LLM-proposed features including epsilon * T, epsilon ** 3, epsilon ** 2, epsilon - T, etc.
>
> Given the statistical context, the LLM reasoned that
>
> “From the dataset and the extracted information, we can observe several important trends and correlations. The output `sigma` (Stress) shows a strong correlation with both `epsilon` (Strain) and `T` (Temperature), with a correlation coefficient of 0.693 for both. The R² values also support this, indicating that both variables contribute significantly to predicting Stress. The strongest correlation with `sigma` is observed with `epsilon_squared`, followed by `epsilon_cubed`, and `epsilon_minus_T`. Additionally, the R² values for `sigma_vs_epsilon_squared` and `sigma_vs_T_squared` suggest that non-linear relationships may be more appropriate than linear ones. Given the high correlation with `epsilon_squared` and the fact that Stress is often related to the square of Strain in physical models (such as in Hooke's Law or stress-strain relationships), we can infer that the function might be a quadratic or cubic function of Strain, possibly combined with Temperature.”,
>
> which leads to the discovered equation “output = params[0] * epsilon + params[1] * T + params[2] * epsilon ** 3 + params[3] * epsilon ** 3 * T” after several search iterations.
>
> The discovered equation, with NMSE of 8.029e-14, closely matches the ground truth, which is “4.255830478818137 * epsilon ** 3 - 0.39029568781474044 * (T - 308.8646781296164) + epsilon ** 3 * 8.63802815432207 * (T - 308.8646781296164)”, demonstrating how statistical context guides the LLM toward physically meaningful functional forms.
>
> We have revised our manuscript with a section that contains this and other qualitative examples in Section D in the Appendix.
>
> Regarding ablations, we **did** compare against a **fixed, hand-written analysis baseline** named "Statistical Hint" (Table 1 in the paper). ProAug significantly outperforms this baseline—for example, on LSR-Transform, Statistical Hint achieves NMSE of 0.188 while ProAug achieves 0.067. This demonstrates that LLM-generated statistical feature analysis provides substantial benefits over predetermined hand-written scripts.

---

> > ### Author Response · Authors · 2025-11-28
> >
> > **Regarding Weakness 1 & Question 2 (Model Size Dependence):**
> > Our experiments span multiple model scales: 4B and 8B parameter models, as well as larger closed-source models. This demonstrates applicability across different model sizes. Furthermore, as LLM capabilities continue to improve and efficiency increases, we expect ProAug's applicability to broaden. We acknowledge that smaller models may face challenges with extended context, but our results show that even small-sized models (4B, 8B parameters) benefit substantially from our approach.
> >
> > **Regarding Weakness 2 & Question 3 (Computational and Engineering Cost):**
> > Regarding execution time, accurate comparison is challenging due to significant fluctuations in API response times across different periods and the substantial influence of external factors. So we conducted a statistical comparison of token consumption between the ProAug and LLMSR algorithms on the Deepseek platform. The results indicate that ProAug consumes approximately 3307 tokens per round on average, while LLMSR consumes about 1327 tokens per round. Although ProAug incurs higher computational costs, these remain within an acceptable range. With ongoing optimization of the baseline model, the associated computational overhead is expected to decrease further. Additionally, we evaluated the environment error rate during code execution. It was observed that the code generated by Deepseek, demonstrates high stability in our tasks, with errors occurring infrequently and an error rate below 10%.
> >
> > **Regarding Weakness 3 & Question 4 (Benchmark Coverage):**
> > The 33 tasks used in our experiments were selected **randomly** from LLM-SRBench to ensure unbiased evaluation. This represents a substantially larger subset than comparable work (e.g., other papers such as LLM-SR use fewer than 10 tasks), strengthening our generalizability claims. We selected this task sample size to balance comprehensive evaluation with computational feasibility, and our random sampling approach provides strong evidence of broad applicability across diverse problem types.
> >
> >
> >
> > **Regarding Weakness 5: (Noisy and Real-World Data):**
> > We respectfully note that LLM-SRBench, proposed recently, represents the current state-of-the-art LLM SR benchmark, specifically designed to approximate real-world discovery scenarios. Unlike traditional benchmarks that rely on well-known equations vulnerable to LLM memorization, LLM-SRBench incorporates diverse problem structures more representative of practical scientific tasks. We agree that evaluation on noisy datasets is a promising direction, one that remains largely unexplored in the current literature. Our framework is well-positioned for such extensions: while traditional LLM-based methods typically depend solely on MSE signals that can become unreliable under noise, our programmatic augmentation can incorporate robust statistical tools specifically designed to maintain robustness in noisy settings. This represents a valuable direction for future work.
> >
> > Thank you for your thoughtful guidance in improving our manuscript. We hope our responses have adequately addressed the points you raised and welcome any further discussion. We are committed to incorporating your valuable feedback to strengthen our paper.

---

### Official Review · Reviewer_YAXf · 2025-10-28

**Soundness:** 3
**Presentation:** 2
**Contribution:** 3
**Rating:** 4
**Confidence:** 5

**Summary:**

This paper proposes PROAUG, a framework that extends LLM-based symbolic regression by introducing a programmatic data-analysis phase. Instead of using only scalar loss signals, the LLM generates and executes Python programs to compute dataset statistics, which are then used as contextual input for equation generation. Experiments on LLM-SRBench show notable reductions in NMSE compared to LLM-SR and statistical-hint baselines.

**Strengths:**

The main advantage is that the idea of using additional information to guide equation evolution is interesting and potentially useful.

**Weaknesses:**

The results are not fully consistent with LLM-SRBench, and the ablation studies are insufficient.

**Questions:**

1. The reported NMSE values differ significantly from those in the original LLM-SRBench paper. Although different foundation models are used, some baseline models (e.g., Llama3.1 in Biology) achieve NMSE around 1e-6, much better than the values shown here. This raises concerns about evaluation consistency.
2. Section 3 presents a case where statistical hints help LLM-SR, but it mainly reflects a local search behavior. The authors should include a concrete example on LSR-Syn to demonstrate how GPT-generated statistical information specifically improves LLM-SR performance.
3. The proposed method does not use the same prompt as LLM-SR. An ablation is needed to confirm whether improvements come from the proposed augmentation or simply from a new prompt design. In particular, Appendix C.2 uses a linear regression model as a seed for PROAUG, which might bias the results.
4. The computational cost is not reported. The paper should clarify how much additional time is required for code generation and data analysis.

---

> ### Author Response · Authors · 2025-11-24
>
> We sincerely thank the reviewer for the thoughtful feedback and constructive comments. We address your questions as follows:
>
> ### Question 1: Differences in reported experiment values
>
> The discrepancy in results stems from different experimental settings rather than evaluation inconsistency. LLM-SRBench reports results for 1000 iterations, whereas we use 150 iterations due to computational budget constraints (running 1000 iterations across all problems of LLM-SRBench with our models would cost thousands of dollars per experimental round). Importantly, Figure 1 in LLM-SRBench shows results at comparable iteration counts that align closely with our reported values.
>
> This issue is compounded by the high sensitivity to randomness inherent in LLM-based SR methods, which we analyze in Section 4.4.2. We find that LLM-based SR methods can have high sensitivity to randomness, which highlights a critical gap in current LLM-based SR research, where multi-run evaluations are rarely reported (including in LLM-SRBench). We recommend that future studies adopt repeated trials to mitigate randomness and enhance reliability.
>
>
> ### Question 2: Concrete example from LSR-Synth
>
> We provide a concrete example demonstrating how GPT-generated statistical information improves performance on MatSci9 from the Material Science category in LSR-Synth. The problem is about finding the equation that represents Stress (sigma), given data on Strain (epsilon), and Temperature (T).
>
> ProAug generates code to extract statistical information including correlation and regression r2 values for LLM-proposed features including epsilon * T, epsilon ** 3, epsilon ** 2, epsilon - T, etc.
>
> Given statistical context, the LLM reasoned that
>
> “From the dataset and the extracted information, we can observe several important trends and correlations. The output `sigma` (Stress) shows a strong correlation with both `epsilon` (Strain) and `T` (Temperature), with a correlation coefficient of 0.693 for both. The R² values also support this, indicating that both variables contribute significantly to predicting Stress. The strongest correlation with `sigma` is observed with `epsilon_squared`, followed by `epsilon_cubed`, and `epsilon_minus_T`. Additionally, the R² values for `sigma_vs_epsilon_squared` and `sigma_vs_T_squared` suggest that non-linear relationships may be more appropriate than linear ones. Given the high correlation with `epsilon_squared` and the fact that Stress is often related to the square of Strain in physical models (such as in Hooke's Law or stress-strain relationships), we can infer that the function might be a quadratic or cubic function of Strain, possibly combined with Temperature.”,
>
> which leads to the discovered equation “output = params[0] * epsilon + params[1] * T + params[2] * epsilon ** 3 + params[3] * epsilon ** 3 * T” after several search iterations.
>
> The discovered equation, with a NMSE of 8.029e-14, closely matches the ground truth, which is “4.255830478818137 * epsilon ** 3 - 0.39029568781474044 * (T - 308.8646781296164) + epsilon ** 3 * 8.63802815432207 * (T - 308.8646781296164)”, demonstrating how statistical context guides the LLM toward physically meaningful functional forms.
>
> We have also revised our manuscript with a section that contains this and other qualitative examples in Section D in the Appendix. Thank you for the suggestion!
>
>
> ### Question 3: Prompt design
>
> We clarify that aside from the  statistical information augmentation component, we share the same prompt structure as the original LLM-SR/LLM-SRBench paper. Importantly, both the original LLM-SR prompt and our method use a linear regression model as the seed hypothesis at the beginning of the search process—this is not unique to ProAug. The difference is the addition of programmatically-generated statistical context component.
>
> We have also revised our manuscript with a section that contains these discussions in Section E in the Appendix.
>
>
> ### Question 4: Computational cost
> The main additional overhead is that ProAug requires two LLM calls per iteration (one for statistical code generation, one for equation generation) versus one call for original LLM-SR. However, this can be viewed as a single tool-use process,  where the LLM generates analysis tools (statistical code), obtains tool-call results (extracted statistics), and performs the final action (equation generation). As LLM inference techniques continue to advance and become more efficient, we expect this overhead to diminish further. We have also revised our manuscript with a section that contains this discussion in Section E in the Appendix.
>
> We are grateful for your thoughtful guidance in improving our manuscript. We hope our responses have adequately addressed the points you raised and are committed to incorporating your valuable feedback to strengthen our paper.

---

### Official Review · Reviewer_cZSc · 2025-10-30

**Soundness:** 1
**Presentation:** 2
**Contribution:** 1
**Rating:** 2
**Confidence:** 5

**Summary:**

This work introduces ProAug for LLM-based symbolic regression which allows LLMs to produce code that is then executed on the dataset to get statistics which is then fed back into the LLM. ProAug is then somewhat evaluated on LLM-SRBench, albeit not adopting the benchmark practices/results.

**Strengths:**

The introduction of a mechanism that allows LLMs to produce code that is then executed on the dataset to get statistics is novel for LLM-based SR.

**Weaknesses:**

Cites but lacks critical discussion or analysis with highly relevant work. A quick search on LLM-based SR methods yields RAG-SR [1] from ICLR 2025, which the paper cites briefly “These methods train networks on experimental data to either directly….or to accelerate the search for target expressions.”. Insufficient discussion and comparison are made to RAG-SR. The critical reason why RAG-SR caught my attention is because this paper claims that (and I quote from the introduction) “existing LLM-based approaches typically rely on scalar evaluation metrics, such as mean squared error, as the sole source of feedback during the search process, thereby overlooking the rich information embedded in the dataset”, yet RAG-SR addresses this exactly, with “semantics” (referring to vector of errors, instead of a scalar. The lack of discussion and empirical comparison to highly relevant work, especially from the same conference, is concerning.

[1] Zhang, Hengzhe, et al. "RAG-SR: Retrieval-augmented generation for neural symbolic regression.", ICLR 2025

Results (see Table 1) are inconsistent with literature, with errors several orders of magnitude higher (except “Material Science” category), even in the original LLM-SRBench paper [2]. Please see Table 1 in [2], and look at the LLM-SR results. In fact LLM-SRBench benchmarks on 3 other competitors, Direct Prompting, SGA, LaSR, which are not considered in this paper. Even LaSR beats most results that this paper presents, by orders of magnitude, which brings into questions the effectiveness of the algorithm.

[2] Shojaee, Parshin, et al. "LLM-SRBench: A new benchmark for scientific equation discovery with large language models." ICML 2025

Another point is that since the paper is using the experiments and standards borrowed from LLM-SRBench, the symbolic accuracy (SA) and numeric precision (Acc_0.1) should be reported too. LLM-SRBench’s value lies in its completeness and transparency, yet the current results include only a subset of algorithms and a single metric. I encourage the authors to include all benchmarked algorithms and multiple metrics to provide a fair, reproducible, and fully informative comparison.

Lack of details to reproduce experiment. Without additional details, I can only assess that prompts seem heavily reliant on the human designer. There is only one example in the Appendix as well, which makes it difficult to assess the approach.

Minor point, please use algorithm names consistently, e.g., “LLMSR”, “LLM-SR” are used interchangeably in the paper.

Finally, paper lacks confidence intervals, error bars and statistical tests for the results. It is hard to determine the significance of the results, even the issues above are resolved.

**Questions:**

Ironically, although the work is about symbolic regression, which discovers equations, I am unable to find an example of the equation that ProAug has discovered. Can the paper provide the equations that are discovered, or give a few clear examples of the f(x) discovered (alongside their metrics)?

How are the prompts developed? Are they ad-hoc or did they undergo a systematic process?

Were the prompts vetted or proofread? There are some grammatical and spelling mistakes in the Appendix, were these the actual prompts or is this just a typographical error in the paper?

Please address the weaknesses listed above as well.

---

> ### Author Response · Authors · 2025-11-24
>
> We sincerely thank the reviewer for the valuable and detailed feedback. We address each concern below:
> ### Weakness
>
> **Discussion of RAG-SR:** We appreciate the opportunity to clarify the fundamental differences between RAG-SR and our method. Our work focuses on improving **LLM-based SR methods** that leverage the physical reasoning and world knowledge capabilities of **large** language models as agents (e.g., LLM-SR, LaSR). In contrast, RAG-SR employs transformer-based neural networks **without** LLM-scale world knowledge or physical reasoning components, representing a fundamentally different methodological paradigm. Our methodology of leveraging programmatic interaction with the dataset to extract complex statistical insights for enhancing reasoning and search is also fundamentally different from RAG-SR.
>
> That said, we acknowledge that RAG-SR reconciles with our motivation that scalar feedback can be limited. We agree that these approaches complementarily work on different methodological lines and have expanded our discussion of RAG-SR in the related work in Section 5 to clarify these distinctions and connections.
>
> **Differences in experimental results:** The discrepancy in results stems from different experimental settings. LLM-SRBench reports results for 1000 iterations, whereas we use 150 iterations due to computational budget constraints (1000 iterations across all problems of LLM-SRBench would cost thousands of dollars per experimental round in our setting). Importantly, Figure 1 in LLM-SRBench shows results at similar iteration counts that align with our reported scale.
>
> This issue is compounded by the high sensitivity to randomness inherent in LLM-based SR methods, which we analyze in Section 4.4.2 and also describe in the “Statistical confidence” point below.
>
> **Additional baselines:** We will include results for more baselines in the paper. Below are preliminary results for the requested baselines on LSR-Transform:
>
> Direct Prompting (No data-driven reasoning, 2 equations sampled from initial prompt): 0.2420
>
> SGA: 1.78367
>
> LaSR (15% of original iterations used in LLMSR-Bench to match our 150-iteration LLMSR setting): 0.2106
>
> In comparison, ProAug achieves an NMSE of 0.067 in this setting.
>
> **Additional metrics:** Symbolic accuracy evaluation relies on LLM-based assessment and requires specific prompting strategies. Since the original evaluation code is not publicly available from LLM-SRBench, ensuring fair comparison is challenging. We have added numeric precision (Acc_0.1) results in Section C in the Appendix.
>
> **Reproducibility and Prompt details:** For both LLM-SR and ProAug, experiments for all instances use the same prompt template, provided in Section E.1 in the Appendix. ProAug shares the same prompt structure as LLM-SR, aside from the statistical information augmentation component. For both LLM-SR and ProAug, the prompts for different problem instances are generated by replacing the input and output variable names and short descriptions in the template with instance-specific values. We have revised our paper with a section that contains these discussions in Section E in the Appendix.
>
>
>
> **Algorithm naming:** Thank you for catching this inconsistency. We have standardized the naming to "LLM-SR" throughout.
>
> **Statistical confidence:** We provide variance analysis in Section 4.4.2, revealing significant sensitivity to randomness in LLM-based SR methods. This highlights a critical gap in current research: multi-run evaluations are rarely reported in prior work (including in LLM-SRBench). We recommend that future studies adopt repeated trials to mitigate randomness and enhance reliability.

---

> > ### Author Response · Authors · 2025-11-24
> >
> > ### Questions
> >
> > **Equation examples discovered:** Thank you for this important question. Below are several equation examples that ProAug discovered.
> >
> >
> > Example 1: I.32.5_1_1
> >
> > Problem background: radiation from accelerating charges in classical electrodynamics.
> >
> > Ground truth:  sqrt(6)\*sqrt(pi)\*sqrt(Pwr\*c\*\*3\*epsilon)/q
> >
> > NMSE: 3.15e-14
> >
> > Symbol meaning: [Target: 'the acceleration of the charged particle', Feature: 'the power of an electromagnetic wave', 'the charge of a particle', 'the electric constant or permittivity of free space', 'the speed of light']
> > ```
> > def equation(Pwr: np.ndarray, q: np.ndarray, epsilon: np.ndarray, c: np.ndarray, params: np.ndarray) -> np.ndarray:
> >     term1 = np.power(Pwr, params[0])
> >     term2 = np.power(q, params[1])
> >     term3 = np.power(epsilon, params[2])
> >     term4 = np.power(c, params[3])
> >     output = params[4] * term1 * term2 * term3 * term4 + params[5]
> >     return output
> > ```
> >
> > Example 2: I.37.4_0_1
> >
> > Problem background:  interference and intensity relations for two coherent wave sources.
> >
> > Ground truth: 2\*I2\*cos(delta)\*\*2 + I2 + Int + 2\*sqrt(I2\*(I2\*cos(delta)\*\*2 + I2 + Int))\*cos(delta)
> >
> > NMSE: 2.68e-14
> >
> > Symbol meaning: [Target: 'the intensity of the first wave source', Feature: 'the resultant intensity of two wave sources', 'the intensity of the second wave source', 'the phase difference between the two wave sources']
> >
> > ```
> > def equation(Int: np.ndarray, I2: np.ndarray, delta: np.ndarray, params: np.ndarray) -> np.ndarray:
> >     cos_term = np.cos(delta + params[2])
> >     sqrt_I2 = np.sqrt(np.maximum(I2, 1e-8))
> >     a = 1.0
> >     b = params[1] * cos_term * sqrt_I2
> >     c = params[0] * I2 - Int
> >     discriminant = b**2 - 4*a*c
> >     valid_mask = discriminant >= 0
> >     sqrt_I1 = np.zeros_like(Int)
> >     sqrt_I1[valid_mask] = (-b[valid_mask] + np.sqrt(discriminant[valid_mask])) / (2*a)
> >     I1_result = np.maximum(sqrt_I1**2 + params[3] + params[4] * I2 * cos_term, 0)
> >     return I1_result
> > ```
> >
> >
> > Example 3: III.15.27_2_0
> >
> > Problem background: quantized wave modes in periodic physical systems.
> >
> > Ground truth: 2\*pi\*alpha/(k\*n)
> >
> > NMSE: 3.54e-15
> >
> > Symbol meaning: [Target: 'the diameter or a characteristic length', Feature: 'the spring constant or a proportionality constant', 'a material-dependent constant or a dimensionless parameter', 'the number of turns or a dimensionless quantity']
> > ```
> > def equation(k: np.ndarray, alpha: np.ndarray, n: np.ndarray, params: np.ndarray) -> np.ndarray:
> >     output = params[0] * alpha / (k * n)
> >     return output
> > ```
> >
> >
> > These examples demonstrate ProAug's ability to discover accurate equations across diverse problems with low NMSE that closely match the ground truth. We have also revised our manuscript with a section that contains these and additional qualitative examples in Section D in the Appendix. Thank you for the suggestion!
> >
> > **Prompt design process:** Our prompts are designed systematically, with further details provided in Section 3.1. Aside from the statistical information augmentation component, ProAug shares the same prompt structure as LLM-SR to ensure fair comparison. We have also reformatted the presentation of prompts in Section E in the Appendix.
> >
> >
> > **Grammar and spelling:** Thank you for pointing this out. We have reformatted the presentation of prompts in Section E in the Appendix. Please let us know if any issues remain.
> >
> >
> > Thank you for your excellent advice and for helping us improve the paper. We hope this response adequately addresses your concerns. We are committed to incorporating your valuable feedback to strengthen our paper and are happy to address any follow up questions.

---

### Official Review · Reviewer_Jp17 · 2025-10-31

**Soundness:** 3
**Presentation:** 3
**Contribution:** 3
**Rating:** 8
**Confidence:** 3

**Summary:**

This paper presents PROAUG, a framework for enhancing large language model (LLM)-based symbolic regression through programmatic context augmentation. The approach enables the LLM to generate and execute code that performs dataset analysis (e.g., correlations, transformations, descriptive statistics) prior to equation generation. The resulting analysis outputs are inserted back into the LLM prompt, enriching the contextual information available during symbolic search.

**Strengths:**

The paper identifies a clear limitation in existing LLM-based SR systems and proposes a practical solution that emulates human-like exploratory data analysis.

The concept of integrating code-generated statistical insights into prompts is novel and well-aligned with the emerging field of neuro-symbolic and programmatic reasoning.

**Weaknesses:**

The main limitations of this work lie in its restricted autonomy, heuristic guidance, and limited evaluation scope.

* The framework relies on a predefined set of data analysis operations, such as basic statistical measures and simple transformations, e.g., log, exp, sin, cos. This restricts its ability to discover equations that depend on highly complex or non-standard data relationships.


* The correlation metrics used in the statistical context are manually designed and confined to a small set of predefined transformations. This may result in the failure to detect intricate nonlinear or multivariate interactions.


* During supervised fine-tuning, the model occasionally generates non-executable code. While the authors mitigate this issue by augmenting training data with examples of failed code generation, it remains a limitation.

**Questions:**

How do you plan to address the limitation of relying on predefined data analysis operations to enable the discovery of more complex or non-standard data relationships in future iterations of PROAUG?


Given the one-to-many mapping issue in the LSR-Synth dataset subset, do you have plans to adapt supervised fine-tuning (SFT) for such cases, or are there alternative approaches you are considering to improve performance on these tasks?

---

> ### Author Response · Authors · 2025-11-24
>
> We sincerely thank the reviewer for the positive evaluation and thoughtful feedback. We address the questions as follows:
>
> ### Question 1: Predefined data analysis operations
> One of our goals with ProAug is to provide an accessible mechanism for scientists to inject domain knowledge into the equation discovery process. For this purpose, having a fixed, verifiable, and well-understood augmentation template is often more valuable than having an adaptive but uncontrollable one. A fixed template enables scientists to iteratively refine the possible augmentations to suit their domain, facilitating collaborative human-AI discovery.
> Our current set of operations was chosen to enable general purpose augmentations across scientific domains such as physics and chemistry.
>
> That said, we see several promising directions to further enhance expressiveness and adaptivity, particularly for non-standard data relationships:
>
> **1. Operation template diversity:** Rather than relying on a single fixed template, we can maintain a library of diverse operation templates and dynamically adjust their priorities based on dataset characteristics. This would improve both the expressiveness and adaptive coverage of the data analysis component.
>
> **2. Meta-instruction optimization:** We can treat the analysis instructions as learnable meta-knowledge and employ LLM-based evolutionary search or optimization to iteratively refine these instructions, enabling the framework to discover more effective analysis strategies automatically.
>
> ### Question 2: One-to-many mapping issue
> The one-to-many nature of LSR-Synth indeed poses challenges for standard supervised fine-tuning. We see several promising approaches to mitigate this limitation:
>
> **1. Enhanced context augmentation:** By enriching input prompts with high-quality, dataset-specific insights (e.g., detected patterns, key statistics, structural properties), which is also relevant to the Question 1 above, we can create more distinctive prompts for different problems, effectively reducing the severity of the one-to-many mapping.
>
> **2. Multi-solution generation:** Instead of generating a single equation per inference, we can modify the pipeline to output a diverse candidate set containing multiple potential solutions with varied functional forms each time. This approach naturally accommodates the inherent one-to-many nature of LSR-Synth and could improve the likelihood of discovering correct equations.
>
> We thank the reviewer again for the constructive feedback and hope our response adequately addresses these important considerations.

---

### Meta-Review · Area_Chair_TLup · 2026-01-02

**Summary:**

The main concerns of the reviewers are:
* Framework is restrictive, since it is based on predefined set of operations
* Generation of non-executable code.
* Only manually defined correlation metrics
* Inconsistent results in the numerical experiments/evaluation
* Insufficient embedding in current literature
* Insufficient numerical experiments (in partiular, ablation study)
* Not fully consistent with LLM-SRBench
* Problems with reproducibility of the results

**Reviewer Concerns:**

Most concerns were addressed, but some not, in particular:
* Even the one from the most positive reviewer (Reviewer Jp17 with score 8) about "Framework is restrictive, since it is based on predefined set of operations" is not sufficiently addressed. This is a limitation of the approach.

**Reviewer Scores:**

It’s really hard to say how any reviewer would have changed their score if they had taken part more fully in the discussion. Without hearing it from them directly, anything we write here would just be guesswork.

For this paper, the scores are 8,2,4,4. But if I had to guess, due to my comment above, I doubt that the 8 would change. As for the others, the strength they mentioned are so weak (and often they just mentioned only one strength), that I have serious doubt those scores would change, maybe only by one. Hence this is to my mind a reject.

---

### Decision · Program_Chairs · 2026-01-26

Reject